# Use peripheral blood leukocyte parameters combined with inflammatory indicators in diagnosis and severity assessment of *mycoplasma pneumoniae* pneumonia in children

Yuwen Zhang[1], Yuanpeng Zhai[1], Shuai Qi[1‡], Dan Huang[1‡], Jingjing Wang[1,2‡], Linyan Wang[2‡], Xuemei Dong[2‡], Zhipeng Sun[3‡], Jiaojiao Yin[2]*, Chong Zhang[1,2]*, Weikai Wang[2]

**1** School of Public Health, Gansu University of Chinese Medicine, Lanzhou, Gansu, People's Republic of China, **2** Department of Clinical Laboratory, Gansu Provincial Maternity and Child-care Hospital, Lanzhou, Gansu, People's Republic of China, **3** Department of Scientific & Application, Sysmex Shanghai Ltd, Shanghai, China

☯ These authors contributed equally to this work.
‡ SQ, DH, JW, LW, XD, and ZS authors also contributed equally to this work.
* yinjiaojiao87@126.com (JY); zhch1972@163.com (CZ)

## Abstract

### Background

*Mycoplasma pneumoniae* is a common cause of pneumonia in children over 5 years of age. Cell Population Data (CPD) is valuable for screening various diseases, including certain infections and myeloproliferative disorders, but less used in the diagnosis of children *mycoplasma pneumoniae* pneumonia (MPP).

### Aims

To improve the diagnostic efficacy and severity assessment of MPP in children by combining CPD with inflammatory markers.

### Methods

This study was conducted at the Gansu Provincial Maternity and Child-care Hospital, China. The data was accessed on 01.10.2024 for the purpose of research. A retrospective study from 01.10. 2022–01.01.2024 included three groups of children aged 5–9 years: 1) those with MPP (n = 240), of which 73 were mild, 167 were severe; 2) those with bacterial pneumonia (n = 52); and 3) those with health check-ups (n = 340). CPD indicators were measured by Sysmex XN-9000 (Kobe, Japan) hematology analyzer. Inflammatory markers included PCT (procalcitonin), LDH (lactate dehydrogenase), CK-MB (creatine kinase isoenzyme), CRP (c-reactive protein), SAA (serum amyloid A), APTT (activated partial thromboplastin time), FIB (fibrinogen). The

**Data availability statement:** All relevant data are within the manuscript and its Supporting Information files.

**Funding:** This study was supported by the Natural Science Foundation of Gansu Province (22JR5RA716, https://kjt.gansu.gov.cn/), Jiaojiao Yin works as the host of this funder, was responsible for the conceptualization, writing-review and editing of the manuscript. Gansu Provincial Science and Technology Major Special Program Project (22ZD6FA034, https://kjt.gansu.gov.cn/), Weikai Wang works as the host of this funder, had had no role in study design, data collection and analysis, decision to publish, or preparation of manuscript.

**Competing interests:** The authors have declared that no competing interests exist.

differences and correlation of parameters between three groups were analyzed and receiver operating characteristic (ROC) analysis was performed.

## Results

Most of the leukocyte parameter indices showed statistically significant differences between the MPP group and the healthy physical examination group ($p < 0.05$). Compared with bacterial pneumonia group, inflammatory markers were not correlated with the occurrence of MPP ($p > 0.05$), while LY-X and NE-WY were negatively correlated with the occurrence of MPP ($p < 0.05$). ROC analysis showed that the areas under the curve (AUC) for LY-X and NE-WY were 0.707 and 0.736, respectively. Additionally, LY-WY and MO-WZ are positively correlated with the severity of MPP ($p < 0.05$), and the AUC for MO-WZ was 0.709. Among the inflammatory markers, the AUC for SAA was 0.828, and combining SAA with MO-WZ increased the AUC to 0.861 in distinguishing mild from severe MPP.

## Conclusion

Peripheral blood leukocyte parameters may have clinical application value for the early diagnosis and disease progression assessment of MPP.

---

## Introduction

*Mycoplasma*s are the smallest known self-replicating prokaryotes with an extremely small genome size of 580–2,200 kbp. They belong to the genus *mycoplasma* within the family mycoplasmataceae and order mycoplasmatales. Of the 16 species of human *mycoplasma*s, 6 are pathogenic, with *Mycoplasma pneumoniae* being the most important and prevalent [1]. *M. pneumoniae* has been identified as causing up to 40% of community acquired pneumonia (CAP) in individuals over the age of five [2]. The infection rate of *M. pneumoniae* decreased following the COVID-19 pandemic in 2020. However, *M. pneumoniae* infections saw a sudden increase in the latter part of the COVID-19 pandemic in 2023. A pediatric hospital in East China reported a positive PCR test rate for MP infection as high as 50% [3].

The diagnosis and differential diagnosis of *mycoplasma pneumoniae* pneumonia (MPP) primarily depend on aetiological and radiological evidence [4], as it cannot be solely deduced from clinical symptoms and signs. Compared to CAP caused by other pathogens, children with MPP often have normal or only slightly elevated white blood cell counts [5,6], which brings certain difficulties for clinical diagnosis. Currently, the aetiological tests for MPP mainly include antigen-antibody testing, *mycoplasma* culture, and nucleic acid testing.

Antibody testing suffers from a prolonged detection window and slow seroconversion, making it difficult to definitively diagnose the current infection, especially when

IgM, IgG, and IgA can be detected in a single serum sample from Polymerase Chain Reaction (PCR)-positive children with asymptomatic *M. pneumoniae* [7]. Antigen testing is faster but has limited sensitivity [8] and does not always meet clinical expectations. PCR technology is fast and has high sensitivity and specificity[9], but it requires a sophisticated laboratory setup which is not widely available in resource-poor settings. Likewise, *Pneumoniae* culture is not commonly performed due to its high technical requirements. Furthermore, inflammatory markers such as procalcitonin (PCT), C-reactive protein (CRP), and serum amyloid A (SAA) have rapidly developed in recent years. hey have certain advantages in distinguishing viral from bacterial infections, yet these indicators have not met clinical expectations in clearly identifying *M. pneumoniae* infections [10]. Therefore, it is critical to explore new and easily obtainable markers for *M. pneumoniae* infection.

Hematology analysis is one of the most frequently conducted tests for patients with infections, when the body is infected, the changes in peripheral blood cells are not only the single occurrence of white blood cells and the proportional change of various classification counts, but also the generation of immature granulocytes such as rod nuclei and a large number of cell morphological changes, including the appearance of neutrophilic toxic particles, vacuoles and nodules. The cytoplasmic particles of lymphocytes increased, and their volume increased. Monocytes migrated and deformed, and their volume and morphology changed to a certain extent[11]. The Sysmex XN-9000 (Kobe, Japan) hematology analyzer can reflect the size of neutrophils, monocytes, lymphocytes and intracytoplasmic structural complexity by using fluorescence flow cytometry technology, combined with forward scattering (FSC), lateral scattering (SSC) and fluorescence intensity (SFL). Cell Population Data (CPD) parameters have certain value in predicting and differentiating infection [11], which can aid in the auxiliary diagnosis of various diseases, including viral infections [12,13], malaria [14,15], sepsis [16], schistosomiasis [17], and others. In addition, it provides reliable results for low cell counts, reducing the need for manual blood film examination while maintaining appropriate diagnostic sensitivity [18].

Integrating CPD parameters with inflammatory markers enhances the diagnostic efficacy and severity assessment of *mycoplasma pneumoniae* pneumonia (MPP) in children. This comprehensive approach facilitates a detailed evaluation of the immune response and inflammation associated with MPP, thereby enabling more accurate and timely clinical management.

## Materials and methods

### Patient data

This study was approved by the Ethics Committee of Gansu Provincial Maternity and Child-care Hospital and adhered to the ethical guidelines of the Helsinki Declaration. The data was accessed on 01.10.2024 for the purpose of research. The study collected retrospective data from children aged 5–9 years diagnosed with MPP and bacterial pneumonia from 01.10.2022 to 01.01.2024. Data of all patients was accessed through the patients' admission ID number. All included pediatric patients underwent medical imaging examination and multiplex PCR tests on sputum or bronchoalveolar lavage fluid, providing radiographic evidence and etiological evidence for the diagnosis. Out of these, 240 children were diagnosed with MPP by clinicians. 52 children were diagnosed with bacterial pneumonia, the highest detection rate was 52.5% (27/52) for *Haemophilus influenzae*, followed by a detection rate of 27.1% (15/52) for *Staphylococcus aureus*, *Streptococcus pyogenes* 9.6% (5/52), *Escherichia coli* 5.8% (3/52) and *Klebsiella pneumoniae* 3.8% (2/52). None of the samples were found to contain *Chlamydophila pneumoniae*, *Legionella pneumophila*, or *Bordetella pertussis*. The control group consisted of 340 healthy children who underwent a medical check-up at our hospital.

### Inclusion Criteria

The inclusion criteria are: (1) Children aged 5–9 years; (2) Hospitalization at Gansu Provincial Maternity and Child-care Hospital during 01.10. 2022 to 01.01.2024; (3) Positive MP PCR test in sputum/bronchoalveolar lavage fluid samples.

## Exclusion criteria

Exclusion criteria include: (1) children with HIV infections, malignancies, confirmed or suspected tuberculosis, undergoing immunosuppressive therapy, immunodeficiency, severe organ dysfunction, congenital heart disease, chronic lung diseases, and other chronic conditions; (2) children with significant missing medical data; (3) hospital admission <24 hours; (4) Patients uncooperative with the treatment or who discontinued treatment prematurely.

## Study groups

A total of 639 participants were initially divided into three groups: (1) The healthy control group (340 individuals); (2) the bacterial pneumonia group (52 individuals); and (3) the MPP group (240 individuals). Based on the 2023 Children's MPP Diagnosis and Treatment Guidelines [4], MPP patients were further categorized into mild and severe groups. Among the MPP patients, 73 cases were classified as mild, and 167 as severe. The criteria for mild pneumonia included the following: 1) Inconsistent with severe disease; 2) The duration of mild MPP is mostly 7~10 days,with generally favorable outcomes and no residual sequelae. The criteria for severe pneumonia included the following: (1) persistent high fever (≥39°C) for ≥5 days or fever for ≥7 days without a decline in peak temperature; (2) presence of one of the following: wheezing, shortness of breath, difficulty breathing, chest pain, or hemoptysis, which could be related to substantial lesions, plastic bronchitis, asthma attacks, pleural effusion, and pulmonary embolism; (3) extrapulmonary complications; (4) oxygen saturation ($SpO_2$) ≤0.93 while breathing room air at rest; (5) imaging showing any of the following: a) involvement of ≥2/3 of a single lung lobe with uniform and consistent high-density consolidation, or b) high-density consolidation in two or more lung lobes (irrespective of the area involved), possibly accompanied by moderate to large pleural effusion and/or localized bronchiolitis; (6) progressive worsening of clinical symptoms with imaging showing that lesion extent progresses more than 50% within 24–48 hours; (7) significant elevations in CRP, LDH, or D-dimer.

## Study methods

2 mL of peripheral venous blood was collected upon admission in both the MPP group and the bacterial pneumonia group. The control group collected fasting peripheral venous blood during the physical examination. Hematology analysis using the Sysmex XN-9000 (Kobe, Japan) analyzer and supporting reagents are combined for the detection of blood routine CPD parameters are automatically generated. The PCT, LDH, CK-MB index data are from the automatic biochemical immunoassay analyzer-c701 (Roche, USA). The CRP, SAA index data are from the specific protein analyzers PA990pro (Lifotronic, China). The APTT, FIB index data are from the automated solidification analyzer CS-5100 (Kobe, Japan). All operations are carried out in strict accordance with the operating procedures.

## Data analysis

Statistical analysis is performed using IBM SPSS Statistics (version 25.0). Kolmogorov-Smirnov test was used to test the normality of each index. ANOVA test was used for normally distributed quantitative data, the Mann-Whitney U test was used for non-normally distributed quantitative data between the two groups, the Kruskal-Wallis test was used for non-normally distributed quantitative data between the multiple groups. Quantitative data following a normal distribution are expressed as mean ± standard deviation (SD), while non-normally distributed data are expressed as median (interquartile range [IQR]). The Pearson's Chi-squared Test for categorical variables, categorical variables are expressed as numbers (percentage). After the overall significant difference between multiple groups was compared, Bonferroni correction test was used, and p-value < 0.05 after adjusting the significance was considered to be statistically significant. Univariate binary Logistic regression analysis was used to explore the relevant variables of MPP, and variables that demonstrated a univariate binary relationship were input into the multivariate binary Logistic regression model, p-value < 0.05 was considered statistically significant. Variables for inclusion were carefully chosen, given the number of

events available, to ensure parsimony of the final model. Receiver Operating Characteristic (ROC) curves and Area Under the Curve (AUC) are plotted to evaluate the different diagnostic efficiencies of leukocyte parameters and inflammatory markers and their predictive value for clinical outcomes, p-value <0.05 is considered statistically significant.

## Results

### Comparison between the MPP group and the healthy control group

The study found no statistically significant difference in gender between the MPP group and the healthy control group ($p > 0.05$). Regarding leukocyte parameters, there were statistically significant differences ($p < 0.05$) in a range of parameters between the two groups, including WBC, NEUT, LYMPH, MONO, EO, BASO, NE-SSC, LY-Y, MO-WZ, NE-SFL, NE-FSC, LY-X, LY-Z, MO-X, MO-Y, MO-Z, NE-WX, NE-WY, NE-WZ, LY-WX, LY-WY, MO-WX, and MO-WY. Specifically, the levels of WBC, NEUT, MONO, NE-WX, NE-WY, LY-WY, MO-WX and MO-WY were higher in the MPP group compared to the healthy control group. Conversely, levels of LYMPH, MO-Y, MO-Z, NE-WZ and LY-WX were lower in the MPP group compared to the healthy control group, as shown in Table 1.

### Comparison between the MPP group and the bacterial pneumonia group

Through multiplex PCR testing, the positive for MP were excluded, yielding 100% (52/52) pneumonia caused by bacterial infection. The highest positivity rate was 52.5% (27/52) for Haemophilus influenzae, followed by a positivity rate of 27.1% (15/52) for Staphylococcus aureus. None of the samples were found to contain Chlamydophila *pneumoniae*, Legionella pneumophila, or Bordetella pertussis. A random sample of 60 children with MPP and sample of 60 healthy children were matched. There was no statistically significant difference between the MPP group and the Bacterial Pneumonia group in terms of gender ($p > 0.05$). In terms of leukocyte parameters, IG, NE-SFL, LY-Z, MO-Y, NE-WX, NE-WY, MO-WX, MO-WY of the MPP group and the Bacterial Pneumonia group were compared with the healthy control group, and the difference was statistically significant ($P < 0.05$). IG, LY-X, NE-WY of the MPP group were lower than those of the Bacterial Pneumonia group, LY-WX of the MPP group was higher than those of the Bacterial Pneumonia group, and the differences were statistically significant ($P < 0.05$), as shown in Table 2. In terms of inflammatory markers, LDH and CRP of the MPP group and the Bacterial Pneumonia group were compared with the healthy control group, and the difference was statistically significant ($P < 0.05$). FIB of the MPP group was higher than the Bacterial Pneumonia group, and the differences were statistically significant ($P < 0.05$), as shown in Table 3. Binary Logistics regression analysis showed that LY-X (OR 0.84,95%CI,0.712–0.991, P = 0.038) and NE-WY (OR 0.992,95%CI,0.985–1, P = 0.039) were correlated with pneumonia pathogen infection, while all inflammatory markers had no correlation, please see Tables 4 and 5. ROC analysis showed that the leukocyte parameters LY-X and NE-WY had an area under the curve greater than 0.7, indicating better efficacy in predicting children with MPP ($p < 0.05$). This part of the result is shown in Table 6 and Fig 1.

### Comparison of children with mild and severe MPP

There was no significant difference in gender between the mild and severe MPP groups ($P > 0.05$), and there was a statistically significant difference in the length of hospital stay and whether alveolar lavage was performed ($P < 0.001$). In terms of leukocyte parameters, NEUT#, LYMPH#, NE-FSC, LY-X, LY-Z, MO-Y, NE-WX, NE-WY, LY-WY, MO-WX, MO-WZ of the mild group and the severe group were compared with the healthy control group, and the difference was statistically significant ($P < 0.05$). MONO#, HFLC#, MO-WZ of the severe group were higher than those of the mild group, and the differences were statistically significant ($P < 0.05$), as shown in Table 7. In terms of inflammatory parameters, LDH, α-HBDH and FIB of mild group and the severe group were compared with the healthy control group, and the difference was statistically significant ($P < 0.01$). LDH, α-HBDH, CRP, SAA of the severe group were higher than those of the mild group, and the differences were statistically significant ($P < 0.01$), as shown in Table 8. Binary logistics regression was used for correlation

**Table 1. The comparison of leukocyte parameters between the healthy control group and the MPP group.**

| leukocyte parameters | Healthy control group (n = 340) | MPP group (n = 240) | p-value |
|---|---|---|---|
| WBC(10^9/L) | 6.78 (5.72-7.91) | 7.57 (5.80-9.88) | < 0.001 |
| NEUT#(10^9/L) | 2.99 (2.21-3.98) | 4.49 (3.04-6.45) | < 0.001 |
| LYMPH#(10^9/L) | 2.87 (2.39-2.45) | 1.84 (1.33-2.52) | < 0.001 |
| MONO#(10^9/L) | 0.41 (0.34-0.50) | 0.54 (0.37-0.78) | < 0.001 |
| EO#(10^9/L) | 0.13 (0.08-0.22) | 0.09 (0.03-0.22) | < 0.001 |
| BASO#(10^9/L) | 0.03 (0.02-0.05) | 0.03 (0.02-0.04) | < 0.001 |
| HFLC#(10^9/L) | 0.02 (0.01-0.04) | 0.01 (0-0.08) | < 0.001 |
| IG(10^9/L) | 0.01 (0.01-0.02) | 0.03 (0.02-0.06) | 0.975 |
| NE-SSC | 152.00 ± 3.87 | 152.09 ± 6.59 | < 0.001 |
| NE-SFL | 47.75 (45.50-50.20) | 46.13 (49.40-51.70) | 0.001 |
| NE-FSC | 85.00 (82.83-87.38) | 87.15 (84.60-90.18) | < 0.001 |
| LY-X | 76.30 (75-77.4) | 77.90 (75.53-79.60) | < 0.001 |
| LY-Y | 66.62 ± 3.51 | 66.61 ± 4.67 | < 0.001 |
| LY-Z | 56.20 (55.6-56.7) | 57.70 (56.80-58.90) | < 0.001 |
| MO-X | 117.80 (116.70-118.60) | 118.75 (116.93-120.20) | < 0.001 |
| MO-Y | 115.65 (109.63-120.58) | 110.45 (104.20-116.88) | < 0.001 |
| MO-Z | 67.00 (65.40-68.90) | 64.45 (62.50-66.40) | < 0.001 |
| NE-WX | 297 (288-305) | 312 (300-328) | < 0.001 |
| NE-WY | 587 (570-607) | 642.5 (612-670) | < 0.001 |
| NE-WZ | 668 (619-707) | 643 (615-677) | < 0.001 |
| LY-WX | 559.5 (530-599) | 540 (501.5-619.8) | 0.021 |
| LY-WY | 912 (871-955) | 953.5 (902-1022) | < 0.001 |
| LY-WZ | 574.5 (500-611) | 543.5 (512-579) | 0.106 |
| MO-WX | 250 (234-265) | 265 (249-282) | < 0.001 |
| MO-WY | 642.5 (589-693) | 691 (623-784) | < 0.001 |
| MO-WZ | 663.57 ± 52.36 | 650.22 ± 51.54 | 0.002 |

*Note*: WBC: White Blood Cell Count; NEUT: Neutrophil absolute concentration; LYMPH: Lymphocyte absolute concentration; MONO: Monocyte absolute Concentration; EO: Eosinophil absolute Concentration; BASO: Basophil absolute Concentration; HFLC: High Fluorescence Large Cell absolute Concentration; IG: Immature granulocyte absolute concentration; NE-SSC: Mean side scattered light distribution width of the neutrophil; NE-SFL: Mean fluorescent light distribution width of the neutrophil; NE-FSC: Mean forward scattered light distribution width of the neutrophil; LY-X: Mean side scattered light intensity of the lymphocyte; LY-Y: Mean fluorescent light intensity of the lymphocyte; LY-Z: Mean forward scattered light intensity of the lymphocyte; MO-X: Mean side scattered light intensity of the monocyte; MO-Y: Mean fluorescent light intensity of the monocyte; MO-Z: Mean forward scattered light intensity of the monocyte; NE-WX: Side scattered light distribution width of the neutrophil; NE-WY: Fluorescent light distribution width of the neutrophil; NE-WZ: Forward scattered light distribution width of the neutrophil; LY-WX: Side scattered light intensity of the lymphocyte; LY-WY: Fluorescent scattered light intensity of the lymphocyte; LY-WZ: Forward scattered light intensity of the lymphocyte; MO-WX: Side scattered light distribution width of the monocyte; MO-WY: Mean Fluorescent light distribution width of the monocyte; MO-WZ: Forward scattered light distribution width of the monocyte.

analysis, in which LY-WY (OR 1.004, 95% CI, 1.000–1.007, P = 0.030) and MO-WZ (OR 1.008, 95% CI, 1.002–1.014, P = 0.010) were associated with MPP severity, please see Table 9. In terms of inflammatory parameters, LDH (OR 1.005, 95% CI, 1.002–1.008, P = 0.002), α-HBDH (OR 1.006, 95% CI, 1.002–1.009, P = 0.003), SAA (OR 1.030, 95% CI, 1.017–1.043, P < 0.001); CRP (OR 1.028, 95% CI, 1.009–1.048, P = 0.004), FIB (OR 1.549, 95% CI, 1.055–2.274, P = 0.025) were associated with MPP severity, with FIB having the highest correlation, as shown in Table 10. ROC analysis showed that the AUC for SAA was the largest (0.828) in inflammatory parameters and the AUC for MO-WZ was the largest (0.709) in leukocyte parameters. We combined these two measures with superior efficacy for predicting severity in children with MPP, with an AUC of 0.861after combination, which has high predictive value. This part of the result is shown in Table 11 and Fig 2.

**Table 2. Comparison of leukocyte parameters in the MPP group, the bacterial pneumonia group and healthy control group.**

| Leukocyte parameters | MPP group (n = 60) | Bacterial Pneumonia group (n = 52) | Healthy control group (n = 60) | p-value |
|---|---|---|---|---|
| WBC (10^9/L) | 7.625(5.86–8.92) | 8.17(5.58–10.97) | 7.25 (6.40–8.52) | – |
| NEUT#(10^9/L) | 4.655(3.07–6.46) | 4.85 (3.12–7.84) | 3.10 (2.35–4.30) | <0.01[bc] |
| LYMPH#(10^9/L) | 1.905(1.39–2.69) | 1.95(1.36–2.71) | 3.12(2.68–3.65) | <0.01[bc] |
| MONO#(10^9/L) | 0.5(0.36–0.63) | 0.515(0.33–0.79) | 0.45(0.35–0.61) | – |
| EO#(10^9/L) | 0.1 (0.05–0.2) | 0.08(0.02–0.22) | 0.16(0.10–0.26) | <0.01[bc] |
| BASO#(10^9/L) | 0.025(0.02–0.038) | 0.03(0.02–0.05) | 0.03(0.02–0.05) | – |
| HFLC#(10^9/L) | 0.03(0.013–0.078) | 0.04 (0.01–0.15) | 0.02(0.01–0.04) | <0.01[bc] |
| IG (10^9/L) | 0.03(0.01–0.07) | 0.04(0.01–0.15) | 0.01(0.01–0.02) | <0.01[bc] |
| NE-SSC | 150.4 ± 4.28 | 151.85 ± 4.53 | 151.00 ± 2.57 | – |
| NE-SFL | 49.05 (46.3–51.225) | 46.7(43.725–48.8) | 48.00(45.70–50.40) | <0.01[bc] |
| NE-FSC | 86.85(83.975–89.725) | 85.7(83.775–89.3) | 84.25(82.83–86.38) | – |
| LY-X | 77.05(75.1–78.775) | 78.45(77.8–79.475) | 75.30(74.90–76.90) | <0.01[abc] |
| LY-Y | 66.86 ± 4.02 | 66.59 ± 5.72 | 67.62 ± 3.51 | – |
| LY-Z | 57.65(56.5–59) | 57.5(56.25–59.3) | 56.20 (55.6–56.7) | <0.01[bc] |
| MO-X | 118.5(117.33–119.73) | 117.75(116–119.45) | 117.55(116.70–118.60) | – |
| MO-Y | 110.05(104.93–115.48) | 111.2(101.825–117.55) | 116.85(111.63–121.58) | <0.01[bc] |
| MO-Z | 64.35(62.4–66.475) | 64.9 (63.225–66.475) | 66.00 (65.40–68.90) | <0.01[bc] |
| NE-WX | 313 (301.25–331) | 315(307.25–338) | 297(285–304) | <0.01[bc] |
| NE-WY | 633(605–651) | 660(631.75–694.75) | 582(565–600) | <0.01[abc] |
| NE-WZ | 637(612–675.25) | 650.5(618–685.5) | 654(616–703) | – |
| LY-WX | 560.5(508.25–627) | 527(495.25–594) | 585.5(530–616) | <0.05[ab] |
| LY-WY | 981(905.25–1041.75) | 960(882.5–1014.75) | 912(871–955) | – |
| LY-WZ | 546 (503.75–578) | 558(496.75–600.25) | 574.5(500–611) | – |
| MO-WX | 263(245.25–278) | 264.5(248.25–275.75) | 250 (234–265) | <0.05[bc] |
| MO-WY | 706.5(648.75–782.25) | 685(603–752.75) | 642.5 (589–693) | <0.05[bc] |
| MO-WZ | 608.31 ± 71.46 | 605.21 ± 71.26 | 663.57 ± 52.36 | – |

*Note*: Difference statistically significant (p < 0.05) after Bonferroni's correction. [a] significant difference between the MPP and the Bacterial Pneumonia groups in post hoc comparison; [b]significant difference between the Bacterial Pneumonia and healthy control groups in post hoc comparison; [c]significant difference between the MPP and healthy control groups in post hoc comparison.

## Discussion

Following the COVID-19 epidemic, the delayed emergence of MP, particularly the incidence of MPP in children continues to rise, causing widespread concern among clinicians [3,19–21]. MP is an important pathogenic factor of community-acquired pneumonia in children, which can trigger 20% to 40% of community-acquired pneumonia during epidemics, and the mortality rate is 1.38% [22]. Moreover, the prognosis of children with severe MPP is usually poor, and complications such as pleural effusion, bronchiectasis, and obliterative tracheitis may occur, which have a serious impact on children's growth and development. Therefore, in clinical practice, early identification and treatment of MPP is of great significance, and early intervention can effectively stop the deterioration of the disease and have a positive impact on the prognosis of children. This study provides a reference for clinical work by studying the differences in leukocyte parameters and other inflammatory indicators and their correlation with the disease in each group of children. Related literature [18–20] has shown that when the body is infected, the changes of peripheral blood cells are not only the single occurrence of WBC and the proportional change of various counts, but also the generation of rod-shaped nuclei and other immature granulocytes and the morphological changes of numerous cells, including the appearance of neutrophils toxic particles, vacuoles, and dule bodies. Tecytoplasmic particles of lymphocytes increased, and their volume increased. Monocytes

**Table 3. Comparison of inflammatory indicators in the MPP group, the bacterial pneumonia group and the healthy control group.**

| Inflammatory indicators | MPP group (n = 60) | Bacterial pneumonia group (n = 52) | Healthy control group (n = 60) | P-value |
|---|---|---|---|---|
| PCT (ng/mL) | 0.23 (0.19–0.30) | 0.30 (0.20–0.45) | 0.28 (0.19–0.46) | – |
| LDH(U/L) | 338 (264–424) | 289 (240–387) | 204 (169–230) | <0.05[bc] |
| CK–MB(U/L) | 24.3 (19.08–37.88) | 29.80 (23.40–44.46) | 24.3 (19.08–35.98) | – |
| α-HBDH(U/L) | 248.00 (201.25–334.25) | 229.00 (193.00–315.00) | 154.67(134.86–174.76) | <0.01[bc] |
| CRP (mmol/L) | 12.70 (5.11–28.87) | 5.99 (0.72–20.28) | 2.67 (0.37–4.79) | – |
| SAA (mg/mL) | 94.12 (24.76–129.69) | 20.29 (5–83.13) | 5.00 (4.80–12.66) | – |
| D-Dimer(mg/L) | 0.65 (0.37–1.74) | 0.29 (0.15–1.33) | 0.36 (0.17–0.48) | – |
| APTT (sec) | 32.05 (30.05–34.60) | 33.20 (29.48–35.23) | 33.00 (30.78–35.67) | – |
| FIB(g/mL) | 3.80 (3.40–4.70) | 3.47 (2.31–4.10) | 3.15 (2.18–3.84) | <0.05[ac] |

*Note*: Difference statistically significant ($p < 0.05$) after Bonferroni's correction. [a]significant difference between the MPP and the Bacterial Pneumonia groups in post hoc comparison; [b]significant difference between the Bacterial Pneumonia and healthy control groups in post hoc comparison; [c]significant difference between the MPP and healthy control groups in post hoc comparison. Abbreviations: PCT: Procalcitonin; LDH: lactate dehydrogenase; CK-MB: Creatine kinase isoenzyme; CRP: C-reactive protein; SAA: Serum amyloid A; APTT: Activated partial thromboplastin time; FIB: fibrinogen

**Table 4. Correlation between leukocyte parameters indicators with pneumonia pathogen infection.**

| Leukocyte parameters indicators | OR | 95%CI | p-value |
|---|---|---|---|
| HFLC | 11.93 | 0.03-112.38 | 0.404 |
| NE-SFL | 1.043 | 0.978-1.112 | 0.196 |
| LY-X | 0.76 | 0.606-0.942 | 0.013 |
| NE-WY | 0.98 | 0.965-0.995 | 0.008 |

**Table 5. Correlation between inflammatory indicators with pneumonia pathogen infection.**

| Inflammatory indicators | OR | 95%CI | p-value |
|---|---|---|---|
| PCT | 0.24 | 0.022–2.65 | 0.25 |
| CKMB | 1.00 | 0.99–1.01 | 0.85 |
| CRP | 1.01 | 0.99–1.03 | 0.22 |
| SAA | 1.008 | 1–1.015 | 0.51 |
| D-Dimer | 0.948 | 0.836–1.076 | 0.41 |
| FIB | 1.939 | 1.167–3.223 | 0.058 |

**Table 6. Predictive value of white blood cell parameters for pneumonia pathogen infection.**

| Leukocyte Parameters | AUC | 95%CI | Cut-off value | Sensitivity | Specificity | p-value |
|---|---|---|---|---|---|---|
| LY-X | 0.707 | 0.612–0.802 | 75.950 | 0.981 | 0.4 | < 0.001 |
| NE-WY | 0.736 | 0.645–0.827 | 653.5 | 0.558 | 0.8 | < 0.001 |

AUC = area under the receiver operating characteristic curve; CI = confidence interval.

migrate and deform, and their volume and morphology change to some extent. At the same time, through the analysis of the indicators reflecting the left shift in granulocyte, monocye, and lymphocyte morphology and the change in intracytoplasmic structure complexity, it was found that they have a certain value in predicting and differentiating infection.

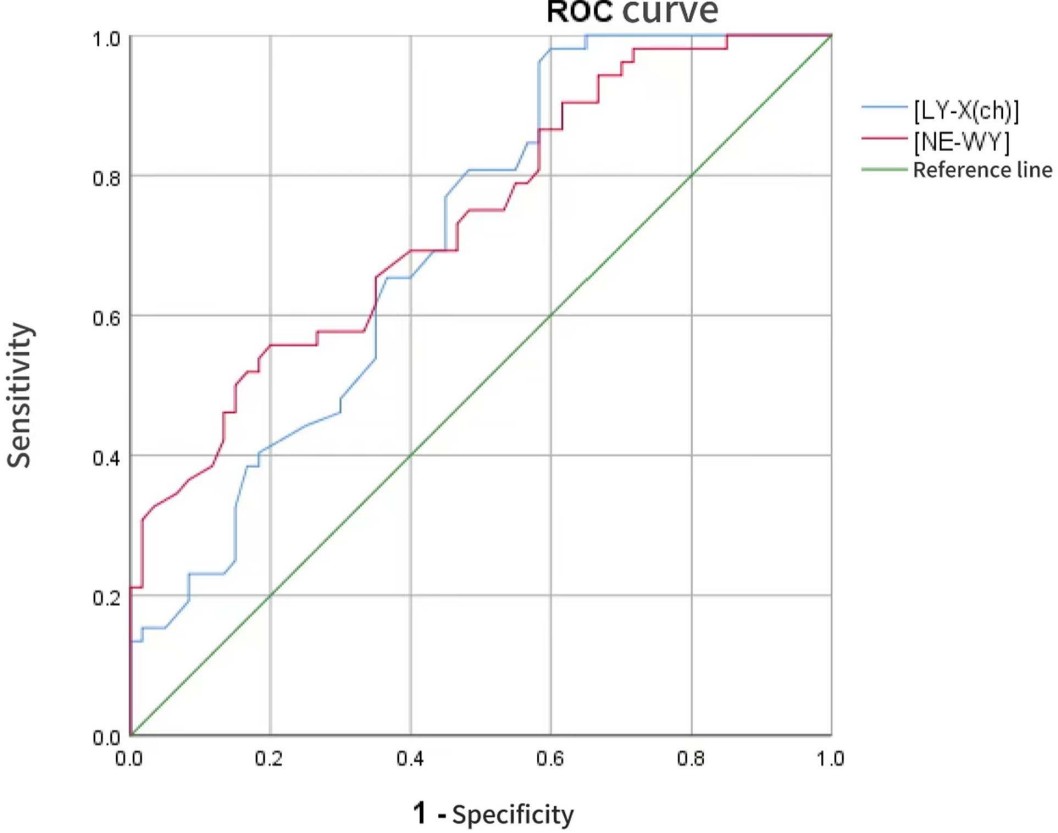

**Fig 1. ROC curve for differential diagnosis of LY-X and NE-WY between the MPP and the bacterial pneumonia.**

Our study has evaluated the CPD reported by the XN Sysmex analyzers. XN Sysmex analyzers also allow their volume and shape, granularity and nucleic acid content to be differentiated. All these cellular characteristics are altered in response to an infectious stimulus. The technology applied is based on fluorescence-flow cytometry using blood-cell membrane surfactant reagents, and different fluorescence dyes specific for staining nucleic acids and proteins. A hematology counter applying this technology and automated-gating is potentially useful for detecting the activation of neutrophils and monocytes in real-time and in an accurate and reproducible way [23]. In the XN analyzers, the leukocyte differential channel discriminates the leukocytes and the signals are plotted in a scatter gram (WDF). The optical signals along X-axis (side scatter) are proportional to the internal complexity; fluorescence along Y-axis correlates with the nucleic acid content, while forward scatter (Z-axis) is related to cell size. Means and SD of values are recorded for each leukocyte subset. The width

of dispersion of the values represents heterogeneous signals.

MPP is the most main CAP in children aged 5 years and older in China, so the study subjects of this study are children aged 5–9 years, who are divided into MPP group and healthy check-ups, respectively. In the past report [24], the researchers found that CRP, SAA, PCT and other inflammatory indicators in children with MPP are significantly higher than those in healthy children. This study further analyzed the differences in various indicators, especially leukocyte parameters, between children with MPP and healthy children. First, in leukocyte parameters, most indicators were different between the two groups. Among them, WBC, NEUT, MONO, NE-WX, NE-WY, LY-WY, MO-WX and MO-WY in the MPP group were higher than those in the health examination group, LYMPH, MO-Y, MO-Z, NE-WZ and LY-WX were lower

**Table 7. Comparison of leukocyte parameters in the mild group, the severe group, and the healthy control group.**

| Leukocyte parameters | Mild group (n=73) | Severe group (n=167) | Healthy control group (n=340) | P value |
|---|---|---|---|---|
| WBC (10^9/L) | 6.97 (5.43-9.62) | 7.65 (5.93-10.16) | 6.78 (5.72-7.91) | – |
| NEUT#(10^9/L) | 4.02 (2.84-6.03) | 4.74 (3.10-6.86) | 2.99 (2.21-3.98) | < 0.05[bc] |
| LYMPH#(10^9/L) | 1.77 (1.29-2.53) | 1.90 (1.38-2.54) | 2.87 (2.39-2.45) | < 0.05[bc] |
| MONO#(10^9/L) | 0.49 (0.36-0.74) | 0.56 (0.40-0.79) | 0.41 (0.34-0.50) | < 0.05[abc] |
| EO#(10^9/L) | 0.08 (0.03-0.19) | 0.11 (0.04-0.25) | 0.13 (0.08-0.22) | – |
| BASO#(10^9/L) | 0.02 (0.01-0.03) | 0.03 (0.02-0.04) | 0.03 (0.02-0.05) | – |
| HFLC#(10^9/L) | 0.01 (0-0.04) | 0.02 (0-0.10) | 0.02 (0.01-0.04) | < 0.05[ac] |
| IG#(10^9/L) | 0.03 (0.01-0.05) | 0.03 (0.02-0.08) | 0.01 (0.01-0.02) | – |
| NE-SSC | 152.50 148.50-156.25) | 151.60 (146.95-156.46) | 152.00 (148.13-155.87) | – |
| NE-SFL | 48.75 (45.8-51.275) | 49.75 (46.3-51.9) | 47.75 (45.50-50.20) | – |
| NE-FSC | 87.65 (85-90.825) | 86.85 (84.6-89.8) | 85.00 (82.83-87.38) | < 0.05[bc] |
| LY-X | 78.25 (75.85-79.60) | 77.75 (75.43-79.28) | 76.30 (75-77.4) | < 0.05[bc] |
| LY-Y | 67.06±4.69 | 66.45±4.68 | 66.62±3.51 | – |
| LY-Z | 57.6 (56.7-58.7) | 57.7 (56.8-59.2) | 56.20 (55.6-56.7) | < 0.05[bc] |
| MO-X | 118.6 (116.7-120.1) | 118.8 (117-120) | 117.80 (116.70-118.60) | – |
| MO-Y | 111.1 (103.8-118.2) | 110.2 (104.1-116.3) | 115.65 (109.63-120.58) | < 0.05[bc] |
| MO-Z | 64.6 (62.7-66.48) | 64.2 (62.2-66.3) | 67.00 (65.40-68.90) | – |
| NE-WX | 309 (298-322) | 315.5 (301-330) | 297 (288-305) | < 0.05[bc] |
| NE-WY | 638 (606-659.5) | 644.5(618.5-675.5) | 587 (570-607) | < 0.01[bc] |
| NE-WZ | 636.5 (615-674.5) | 646.5 (615-679) | 668 (619-707) | – |
| LY-WX | 523 (484.5-575) | 553 (507-628) | 559.5 (530-599) | – |
| LY-WY | 933 (900-980) | 968 (902-1041) | 912 (871-955) | < 0.05[bc] |
| LY-WZ | 541 (507-578) | 545 (513-579) | 574.5 (500-611) | – |
| MO-WX | 259 (242-282) | 266 (251-283) | 250 (234-265) | < 0.05[bc] |
| MO-WY | 696 (616-761) | 688 (623-791.5) | 642.5 (589-693) | < 0.05[bc] |
| MO-WZ | 556.50±82.13 | 623.13±86.67 | 663.57±52.36 | < 0.05[abc] |

**Table 8. Comparison of inflammatory indicators in the mild group, the severe group and the healthy control group.**

| Inflammatory indicators | Mild group (n=73) | Severe group (n=167) | Healthy control group (n=340) | P value |
|---|---|---|---|---|
| PCT(ng/mL) | 0.23 (0.18-0.28) | 0.24 (0.20-0.30) | 0.27 (0.18-0.42) | – |
| LDH(U/L) | 284.50 (253.75-339.25) | 326.50 (266.25-448.00) | 207 (170-228) | <0.01[abc] |
| CK-MB(U/L) | 24.40 (18.0-38.70) | 23.90 (18.50-35.55) | 23.3 (19.08-34.98) | – |
| α-HBDH(U/L) | 221 (199-266) | 249 (206-351) | 156.25 (137.25-174.76) | <0.01[abc] |
| CRP(mmol/L) | 7.39 (2.06-18.47) | 15.09 (6.64-29.48) | 2.43 (0.44-4.53) | <0.01[ac] |
| SAA(mg/mL) | 23.06 (10.52-66.52) | 102.61±28.82 | 5.00(4.80-12.66) | <0.01[ac] |
| D-Dimer(mg/L) | 0.37 (0.30-0.57) | 0.65 (0.37-1.76) | 0.31 (0.17-0.37) | – |
| APTT(second) | 32.08±0.52 | 32.4 (30.2-34.7) | 33.05 (31.65-35.67) | – |
| FIB(g/mL) | 3.88±0.86 | 4.03 (3.53-4.76) | 3.15 (2.18-3.84) | <0.01[bc] |

*Note*: Difference statistically significant (*p<0.05*) after Bonferroni's correction. [a]significant difference between the mild and the severe groups in post hoc comparison; [b]significant difference between the mild and the healthy control groups in post hoc comparison; [c]significant difference between the severe and the healthy control groups in post hoc comparison.

**Table 9. Correlation of leukocyte parameters/inflammatory indicators with MPP severity.**

| Indicators | OR | 95%CI | P value |
|---|---|---|---|
| LY-WY | 1.004 | 1.0–1.007 | 0.03 |
| LY-WX | 1.001 | 0.998-1.003 | 0.598 |
| MO-WZ | 1.004 | 1.002-1.014 | 0.010 |
| BASO | 1.459 | 0.795-2.678 | 0.223 |
| HFLC | 3.897 | 0.02-774.549 | 0.614 |
| IG | 3.171 | 0.523-19.235 | 0.210 |

**Table 10. Correlation of inflammatory indicators with MPP severity.**

| Indicators | OR | 95%CI | P value |
|---|---|---|---|
| LDH | 1.005 | 1.002-1.008 | 0.002 |
| α-HBDH | 1.006 | 1.002-1.009 | 0.003 |
| CRP | 1.028 | 1.009-1.048 | 0.004 |
| SAA | 1.030 | 1.017-1.043 | < 0.001 |
| D-Dimer | 1.867 | 1.115-3.127 | 0.18 |
| FIB | 1.549 | 1.055-2.274 | 0.025 |

**Table 11. ROC curves for the diagnostic value of leukocyte parameters and inflammatory indicators for MPP severity.**

| Indicators | AUC | 95%CI | Cut-off | Sensitivity | Specificity | P value |
|---|---|---|---|---|---|---|
| MO-WZ | 0.709 | 0.613-0.805 | 529 | 0.89 | 0.425 | < 0.001 |
| SAA | 0.828 | 0.758-0.898 | 50.93 | 0.791 | 0.725 | < 0.001 |
| MO-WZ+SAA | 0.861 | 0.799-0.923 | 0.80 | 0.67 | 0.925 | < 0.001 |

AUC = area under the receiver operating characteristic curve; CI = confidence interval.

than those in the health examination group. It can be seen that the degree of heterogeneity of the neutrophil population, the degree of heterogeneity of the monocyte population, and the degree of heterogeneity of the lymphocyte population vary greatly between the two groups.

The important processes of maintaining stability, resisting infection, repairing injury and immune regulation in the body are accomplished by cell death and removing dead cells. In recent years, many studies have shown [25] that blood cells and their derived parameters can be used as new reference indicators of inflammatory response. However, few studies have focused on the diagnostic value of CPD parameters for *M. pneumoniae* in children. The changes of these cells are not only associated with the inflammatory response induced by *mycoplasma* infection, but also with the activated immune response, in which neutrophils, platelets, and lymphocytes have been found to be closely related to *M. pneumoniae* infection. In order to further explore the specific indicators of children with *M. pneumoniae*, this study also analyzed the differences in CPD and inflammatory indicators between children infected with *M. pneumoniae* and children infected with other pathogens, we divided the pneumonia children who had undergone sputum or bronchoalveolar lavage fluid PCR into two groups, that is the MPP group and the bacterial pneumonia group, and analyzed the basic data, blood cell counts, and leukocyte parameters of these two groups, respectively. The results showed that there were differences in age between the two groups, and children infected with *M. pneumoniae* were older than those not infected with *M. pneumoniae*. In this study, leukocyte parameters were also compared, which were not involved in other studies and reports. We found that

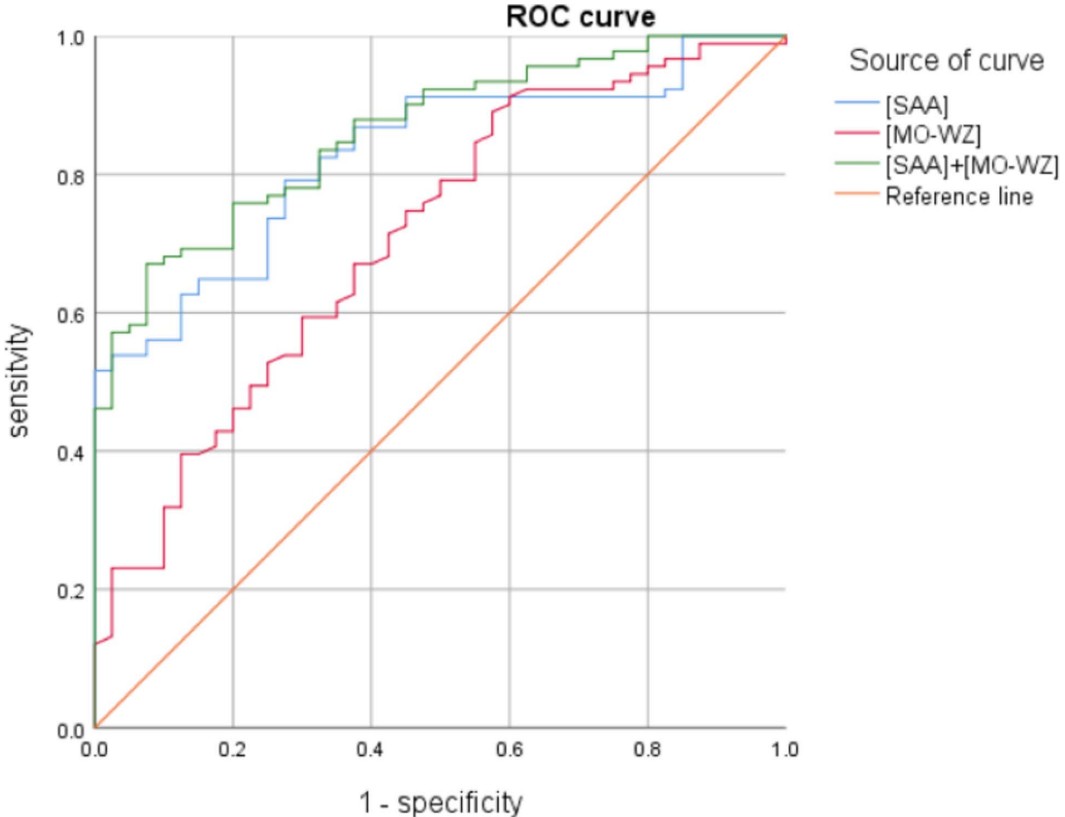

**Fig 2. ROC curve of leukocyte parameters and combined indicators of leukocyte parameters predicting mild and severe MPP in children.**

HFLC, NE-SFL, NE-WY, LY-WX and MO-X in the MPP group were different from those in the bacterial pneumonia group, and HFLC, NE-SFL, LY-WX and MO-X in the MPP group were higher than those in the bacterial pneumonia group, which indicated that the degree of lymphocyte population heterogeneity and monocyte complexity in the MPP group were higher than those in the bacterial pneumonia group, while NE-WY in the MPP group was lower than that in the bacterial pneumonia group, which indicated that the degree of neutrophil population heterogeneity in the bacterial pneumonia group was higher. This may be because neutrophils undergo cell death after reacting to bacteria, including apoptosis and necrosis [26,27], additionally, induce emergency granulopoiesis, which rapidly increases the de novo production of neutrophils. This mechanism results in the presence of both immature neutrophils and mature populations in the peripheral blood, which can act in either an immunosuppressive or pro-inflammatory manner [28,29]. It is reflected in the diversity of nucleic acids and may lead to a higher NE-WY value. Our results also further confirm that NE-WY is associated with pneumonia pathogens.

In general, children with MPP have mild clinical symptoms, and macrolide antibiotics usually respond well to their treatment, however, conditions that have the potential to progress to severe MPP are more common about one week after the onset of the disease. Therefore, it is important to study the risk factors of severe MPP, identify high-risk children early and take corresponding treatment strategies to improve the clinical prognosis of children and reduce the occurrence of severe and sequelae. Many related studies have reported that CRP [30,31], ESR [32,33], PCT [34,35], D-dimer [36,37], LDH [38,39], high *M. pneumoniae* antibody titer [40,41], lower lobe lesions, lobar lesions and fever time are the influencing factors of severe MPP. On this basis, this study conducted a further study on children with mild MPP. First, there were differences in the length of hospital stay and whether alveolar lavage was performed between the mild MPP group and the severe MPP group. In terms of leukocyte parameters, there were significant differences in BASO, HFLC, IG, LY-WX, LY-WY

and MO-WZ between the two groups, and these indicators were lower in the mild MPP group than in the severe MPP group. LY-WX was not associated with the occurrence of mild and severe disease, and LY-WY and MO-WZ were associated with the occurrence of mild and severe disease. Lymphocytes can be divided into a variety of functionally different subgroups,these different cells activation to proliferation, clonal expansion and effector function is crucial for efficient clearance of infection by pathogens [42]. There is heterogeneity in mature monocytes, with the cytoplasm containing a variety of granular structures and diverse nuclear morphology [43]. Meanwhile, Monocyte extravasation and differentiation serve multiple immune functions.The role of monocyte subsets in the resolution of inflammation is increasingly considered [44]. LY-WY and MO-WZ represent the degree of heterogeneity of lymphocyte population and monocyte population,our study further found that the degree of lymphocyte and monocyte heterogeneity was positively correlated with the severity of MPP.

However, this study also has some limitations, (1) this is a retrospective study from a single center, which cannot ensure that children in other regions also have similar characteristics, (2) the number of included cases is small, and the results obtained may be incidental and difficult to perform a detailed subgroup analysis; (3) this study is a retrospective study, and the criteria for including children are demanding and there may be multiple potential biases that are difficult to avoid. Therefore, in future studies, high-quality, multicenter, large-sample studies, as well as prospective studies should be carried out as far as possible to verify and enrich the relevant test indicators related to pediatric MPP. At the same time, expert consensus methods can be used to further validate the results of this study to ensure its accuracy.

## Conclusion

Our results suggest that Leukocyte parameters such as LY-WX and NE-WY have a certain auxiliary role in the diagnosis of MPP, and combining MO-WZ with SAA offers a higher diagnostic value for the severity of MPP. Therefore, peripheral blood leukocyte parameters may have clinical application value for the early diagnosis and disease progression assessment of MPP.

## Supporting information

**S1 Data.**
(XLSX)

**S2 Data.**
(XLSX)

**S3 Data.**
(XLSX)

**S4 Supplementary Figures.**
(ZIP)

**S5 Supplementary related files.**
(ZIP)

## Author contributions

**Conceptualization:** Yuwen Zhang, Yuanpeng Zhai, Jiaojiao Yin.

**Data curation:** Yuwen Zhang, Yuanpeng Zhai, Shuai Qi, Dan Huang, Jingjing Wang, Linyan Wang, Xuemei Dong.

**Formal analysis:** Yuwen Zhang, Yuanpeng Zhai.

**Funding acquisition:** Jiaojiao Yin, Weikai Wang.

**Investigation:** Yuwen Zhang, Shuai Qi, Dan Huang, Jingjing Wang.

**Methodology:** Yuwen Zhang, Yuanpeng Zhai.

**Resources:** Weikai Wang.

**Software:** Yuanpeng Zhai, Zhipeng Sun.

**Supervision:** Chong Zhang.

**Validation:** Yuwen Zhang, Yuanpeng Zhai.

**Visualization:** Zhipeng Sun.

**Writing – original draft:** Yuwen Zhang, Yuanpeng Zhai.

**Writing – review & editing:** Chong Zhang, Jiaojiao Yin.

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
