## [Decision Letter · Decision Letter 0]

9 Aug 2024

PONE-D-24-19765Use peripheral blood leukocyte parameters combined with inflammatory indicators in diagnosis and severity assessment of mycoplasma pneumoniae pneumonia in childrenPLOS ONE

Dear Dr. Zhang,

Thank you for submitting your manuscript to PLOS ONE. After careful consideration, we feel that it has merit but does not fully meet PLOS ONE’s publication criteria as it currently stands. Therefore, we invite you to submit a revised version of the manuscript that addresses the points raised during the review process. Please note that we have only been able to secure a single reviewer to assess your manuscript. We are issuing a decision on your manuscript at this point to prevent further delays in the evaluation of your manuscript. Please be aware that the editor who handles your revised manuscript might find it necessary to invite additional reviewers to assess this work once the revised manuscript is submitted. However, we will aim to proceed on the basis of this single review if possible. 

We look forward to receiving your revised manuscript.

Kind regards,

Jennifer Tucker, PhD

Staff Editor

PLOS ONE

Journal Requirements:

"the Natural Science Foundation of Gansu Province, China (Grant 22JR5RA716)."

"The authors would like to acknowledge the Natural Science Foundation of Gansu Province, China (grant no. 22JR5RA716); Major Project of the Ministry of Science and Technology of Gansu Province, China (grant no. 22ZD6FA034)."

"the Natural Science Foundation of Gansu Province, China (Grant 22JR5RA716)."

6. Please remove your figures from within your manuscript file, leaving only the individual TIFF/EPS image files, uploaded separately. These will be automatically included in the reviewers’ PDF.

Reviewers' comments:

Reviewer's Responses to Questions

**Comments to the Author**

1. Is the manuscript technically sound, and do the data support the conclusions?

Reviewer #1: Partly

2. Has the statistical analysis been performed appropriately and rigorously? 

Reviewer #1: No

3. Have the authors made all data underlying the findings in their manuscript fully available?

Reviewer #1: Yes

4. Is the manuscript presented in an intelligible fashion and written in standard English?

Reviewer #1: Yes

5. Review Comments to the Author

Reviewer #1: The paper presents valuable findings on the use of leukocyte parameters and inflammatory markers to diagnose and assess the severity of MPP in children, but the authors need to address some shortcomings to enhance scientific rigor:

1 How was bacterial pneumonia diagnosed in this study, and which bacterial pathogens were included?

2 Mild MPP is self-limiting, and the diagnostic criteria for severe pneumonia also include disease duration factors.

3 Did the authors consider disease duration factors in the study design, and how did they avoid the influence of disease duration on the study results?

4 The discussion should explain and discuss the significance of leukocyte parameters.

5 The manuscript does not provide sufficient information on whether and how the assumptions for each statistical test were checked. This includes assumptions for normality, homogeneity of variances, and independence.

6 The use of multiple logistic regression models without a clear justification for the selection of variables could lead to overfitting, especially given the relatively small sample size in some groups.

6. PLOS authors have the option to publish the peer review history of their article (what does this mean? ). If published, this will include your full peer review and any attached files.

**Do you want your identity to be public for this peer review?** For information about this choice, including consent withdrawal, please see our Privacy Policy .

Reviewer #1: No

---

## [Author Response · Author response to Decision Letter 1]

25 Sep 2024

1.How was bacterial pneumonia diagnosed in this study, and which bacterial pathogens were included?

The diagnosis of bacterial pneumonia is made by clinicians combining clinical examination with imaging studies to diagnose pneumonia, followed by multiple PCR pathogen detection to rule out mixed infections of viral and bacterial pneumonia, leading to the diagnosis of bacterial pneumonia Among them, the detection rate of Haemophilus influenae was the highest, 52.5% (27/52), followed by Staphylococcus aureus 27.1% (15/52), Streptococcus pyogenes 9.6% (5/52), Escherichia coli 5.8% (3/52) and Klebsiella pneumoniae 3.8% (2/52).

2.Mild MPP is self-limiting, and the diagnostic criteria for severe pneumonia also include disease duration factors.

Regarding the diagnostic criteria for MPP, we reviewed relevant literature The course of mild MPP is generally 7-10 days, while there is no consensus on the course of severe MPP. We found the following about the fever duration in severe MPP: persistent high fever (≥39°C) for ≥5 days or fever for ≥7 days without a decline in peak temperature.

3.Did the authors consider disease duration factors in the study design, and how did they avoid the influence of disease duration on the study results?

Regarding the comparison between MPP and bacterial pneumonia, there was no statistically significant difference in the disease duration between the two groups (P>0.05). However, there was a statistically significant difference in the disease duration between the two groups of mild MPP and severe MPP (P<0.05). We did not consider the disease duration of illness because the disease duration is related to the severity of the disease, and the abnormal diagnostic indicators is also related to the severity of the disease. However, the disease duration may not be directly related to the diagnostic indicators. Furthermore, the severity of the disease determines the disease duration. The disease duration is not a factor that leads to the severity of the disease, so we did not consider the disease duration.

4.The discussion should explain and discuss the significance of leukocyte parameters.

We added the significance of leukocyte parameters in the Introduction and Discussion section.

5.The manuscript does not provide sufficient information on whether and how the assumptions for each statistical test were checked. This includes assumptions for normality, homogeneity of variances, and independence

6.The use of multiple logistic regression models without a clear justification for the selection of variables could lead to overfitting, especially given the relatively small sample size in some groups.

We have supplemented the descriptions related to normality,etc. In this research, statistical approaches were initially employed to determine whether the data conformed to the criteria of normal distribution. The t-test was utilized for quantitative data that adhered to the criteria for normal distribution, whereas the Mann-Whitney U test was applied for non-normal quantitative data. The Pearson's Chi-squared Test was employed for categorical variables. Quantitative data that met the criteria for normal distribution was presented as the mean ± standard deviation (SD), while non-normal distribution data was represented as the median (interquartile range [IQR]). p-value < 0.05 was regarded as statistically significant.

Subsequently, univariate binary Logistic regression analysis was executed, and variables that demonstrated a univariate binary relationship were input into the Multifactorial binary logistic regression, p-value<0.05 was considered statistically significant. Given the number of available events,variables of leukocyte parameters and inflammatory indicators for inclusion were carefully chosen and separately input into the multifactorial binary logistic regression analysis for statistics to avoid overfitting.

Finally, receiver operating characteristic (ROC) curves and the area under the curve (AUC) were plotted to assess the diagnostic efficacy of the white blood cell parameters, p-value <0.05 is considered statistically significant.

---

## [Decision Letter · Decision Letter 1]

3 Jan 2025

PONE-D-24-19765R1Use peripheral blood leukocyte parameters combined with inflammatory indicators in diagnosis and severity assessment of mycoplasma pneumoniae pneumonia in children外周血白细胞参数结合炎症指标对儿童肺炎的诊断和严重程度评价PLOS ONE

Dear Dr. Zhang,

Thank you for submitting your manuscript to PLOS ONE. After careful consideration, we feel that it has merit but does not fully meet PLOS ONE’s publication criteria as it currently stands. Therefore, we invite you to submit a revised version of the manuscript that addresses the points raised during the review process.

We have received the expert reviewer's opinions and invite you to submit a revised version of the manuscript. Please consider and address each of the comments raised by the reviewers.  

We look forward to receiving your revised manuscript.

Kind regards,

Senthilnathan Palaniyandi, Ph.D

Academic Editor

PLOS ONE

Journal Requirements:

Reviewers' comments:

Reviewer's Responses to Questions

**Comments to the Author**

1. If the authors have adequately addressed your comments raised in a previous round of review and you feel that this manuscript is now acceptable for publication, you may indicate that here to bypass the “Comments to the Author” section, enter your conflict of interest statement in the “Confidential to Editor” section, and submit your "Accept" recommendation.

Reviewer #1: (No Response)

2. Is the manuscript technically sound, and do the data support the conclusions?

Reviewer #1: Partly

3. Has the statistical analysis been performed appropriately and rigorously? 

Reviewer #1: No

4. Have the authors made all data underlying the findings in their manuscript fully available?

Reviewer #1: Yes

5. Is the manuscript presented in an intelligible fashion and written in standard English?

Reviewer #1: Yes

6. Review Comments to the Author

Reviewer #1: 1 A large number of leukocyte parameters (e.g., NE-WY, MO-WX) are presented in the results, but the clinical significance of each is not discussed thoroughly.I suggest the authors focus on the most clinically significant parameters. They should discuss in detail how these specific parameters aid in diagnosing MPP or assessing its severity. For example, they should explain why certain parameters, such as MO-WZ, are more predictive of severe MPP than others.

2 The paper appears to conduct multiple comparisons (e.g., comparisons of various leukocyte parameters and inflammatory markers), but there is no mention of adjustments for multiple testing, such as the Bonferroni correction or False Discovery Rate (FDR) control.I recommend applying correction methods to control the increased risk of Type I errors (false positives) when conducting multiple comparisons. The corrected p-values should be clearly indicated in the table, or the authors should explain the reasons for not making corrections.

7. PLOS authors have the option to publish the peer review history of their article (what does this mean? ). If published, this will include your full peer review and any attached files.

**Do you want your identity to be public for this peer review?** For information about this choice, including consent withdrawal, please see our Privacy Policy .

Reviewer #1: **Yes: ** Xu Yongsheng

---

## [Author Response · Author response to Decision Letter 2]

25 Jan 2025

1. A large number of leukocyte parameters (e.g., NE-WY, MO-WX) are presented in the results, but the clinical significance of each is not discussed thoroughly.I suggest the authors focus on the most clinically significant parameters. They should discuss in detail how these specific parameters aid in diagnosing MPP or assessing its severity. For example, they should explain why certain parameters, such as MO-WZ, are more predictive of severe MPP than others.

We have discussed the clinical significance of partially supplemented leucocyte parameters (e.g., NE-WY, MO-WX, etc.) and have highlighted them in red.

2.The paper appears to conduct multiple comparisons (e.g., comparisons of various leukocyte parameters and inflammatory markers), but there is no mention of adjustments for multiple testing, such as the Bonferroni correction or False Discovery Rate (FDR) control.I recommend applying correction methods to control the increased risk of Type I errors (false positives) when conducting multiple comparisons. The corrected p-values should be clearly indicated in the table, or the authors should explain the reasons for not making corrections.

We adjusted the statistical methods as follows:

Kolmogorov-Smirnov test was used to test the normality of each index. ANOVA test was used for normally distributed quantitative data, the Mann-Whitney U test was used for non-normally distributed quantitative data between the two groups, the Kruskal-Wallis test was used for non-normally distributed quantitative data between the multiple groups.After the overall significant difference between multiple groups was compared, Bonferroni correction test was used, and p-value < 0.05 after adjusting the significance was considered to be statistically significant.

---

## [Editor Report · Decision Letter 2]

7 Mar 2025

Use peripheral blood leukocyte parameters combined with inflammatory indicators in diagnosis and severity assessment of mycoplasma pneumoniae pneumonia in children外周血白细胞参数结合炎症指标对儿童肺炎的诊断和严重程度评价

PONE-D-24-19765R2

Dear Dr. Zhang,

We’re pleased to inform you that your manuscript has been judged scientifically suitable for publication and will be formally accepted for publication once it meets all outstanding technical requirements.

Kind regards,

Senthilnathan Palaniyandi, Ph.D

Academic Editor

PLOS ONE
---

## [Editor Report · Acceptance letter]

PONE-D-24-19765R2

PLOS ONE

Dear Dr. Zhang,

I'm pleased to inform you that your manuscript has been deemed suitable for publication in PLOS ONE. Congratulations! Your manuscript is now being handed over to our production team.

Kind regards,

on behalf of

Dr. Senthilnathan Palaniyandi

Academic Editor

PLOS ONE